# Gender Concerns When Noah the Economist Ranks Biodiversity Protection Policies

**David W. Martin** 

Department of Economics and Core Faculty in Environmental Studies, Davidson College, Davidson, NC 28035, USA; damartin@davidson.edu; Tel.: +1-704-894-2264

**Abstract:** Because the funds to protect biodiversity are very limited, biodiversity protection policies are prioritized using the Noah's Ark perspective. I discuss how gender affects Noah's assessment of key elements of his ranking: Discounting, changes in total economic value, marginal costs, changes in ecological value, and the probability of policy success. This incremental approach makes visible the breadth of the mechanisms by which gender differentiated social constructs interact to affect Noah in a richly complex manner.

**Keywords:** biodiversity protection; Noah's Ark; economic valuation; gender differentiation

---

## 1. Introduction

Referring to an iconic set of female images, the farmer sings: "Mother Nature and Mother Earth are two of three women who dictate what I'm worth" (Page and Robertson 1996). A contrasting gender image arises when one thinks of the Ark as important to the Abrahamic religions. According to the *Torah* Book of Bereshit, *Bible* Book of Genesis, and *Qu'ran* Surah 11, God flooded the earth destroying all but the faithful who joined Noah and the species he (!) loaded onto the Ark.

That contrast is important because the Noah's Ark metaphor (Weitzman 1998) is a dominant framework for comparing biodiversity protection policies. It is appealing because the combination of budgetary limitations (McCarthy et al. 2012) and the declining stock of biodiversity (Pereira et al. 2010; Society for Conservation Biology 2019) creates a framework in which policy makers must prioritize what species to save in the same way that the physical limits of the Ark conceivably forced Noah to choose carefully which animals to load. This prioritization subject to a budget constraint is precisely the definition of an economic problem. As most economists are male (Bayer and Rouse 2016) it is quite likely that the real world economic analyst is a male. This fact leads to the simple question that motivates this analysis, does Noah's gender matter?

Building upon Weitzman's (1998) original work, Martin (2016) developed an equation for Noah to use when prioritizing biodiversity protection policies (presented as Equation (1) in Section 2). The basis for this equation is simply a ratio comparing a policy's marginal benefits to its marginal costs. Consistent with basic microeconomic analysis, Noah should first employ the policy with the highest ratio and then continue choosing policies in descending order until reaching the budget constraint. By doing so, Noah will maximize society's welfare in the same way that the Ark was optimally filled with animals. The fundamental question I explore is, does Noah's gender affect the values of the terms used in that equation?

And, of course, Noah's gender matters. As Hawkins et al. state: "*[G]ender matters all the way through* the body, household, home, habitat, city, region, nation, and globe" (Hawkins et al. 2011, p. 243; emphasis in the original). The goal of my analysis flows from Arora-Jonsson:

> And this is perhaps one of the most important contributions of gender scholarship in the past and what it can continue to contribute to environmental studies—making visible

the mechanisms by which environmental governance takes place—the daily practices of knowledge production and action, so as to be able to find openings for better environments but also a more just society. (Arora-Jonsson 2014, p. 12)

I use Equation (1) (in Section 2) to structure the details underlying Noah's valuations so that I can make visible the breadth of mechanisms through which gender plays a role and so that I can pull together different strands of the rich literature in a coherent manner. The net result of my analysis will be discouraging for those who desire a sharp black or white answer, but it is important to look at the individual mechanisms as they interact to build the bigger picture.

In Section 2, I briefly define biodiversity and review how Noah decides which species to save. Section 3 is the heart of this note in which I assess five components of Noah's decision rule. I conclude the note in Section 4 by pulling these elements together.

## 2. The Prioritization of Biodiversity Policies

The starting point for many definitions of biodiversity is E.O. Wilson's:

[A]ll hereditarily based variation at all levels of organization, from the genes within a single local population or species, to the species composing all or part of a local community, and finally to the communities themselves that compose the living parts of the multifarious ecosystems of the world. (Wilson 1996, p. 1)

The neoclassical economic perspective on biodiversity is naturally anthropocentric: it is part of our stock of natural capital, and nature and humans use the flow of its services to produce goods and services we value (Costanza and Daly 1992). Because we value those service flows, economists can value policies that preserve the stock of biodiversity by comparing the lost flow values that would occur if costly protection policies are not undertaken.

Biodiversity is often specified as a public good in that it is both non-excludable and non-rivalrous (e.g., Bezabih and Stage 2019). However, biodiversity is also presented as a club good (e.g., Helm and Hepburn 2012), an open-access good (e.g., Elliott et al. 2019), and as common property (e.g., German 2018). It doesn't matter which of these four types of goods we consider biodiversity to be because the economics of using and paying for each of those types of goods is dependent upon the specific social context. This point importantly foreshadows that social considerations will be a key factor when valuing biodiversity protection instead of relying solely upon market forces.

Martin's (2016) version of Weitzman's (1998) marginal benefits-marginal costs ratio is:

$$R_i = \frac{(\Delta TEV_i)}{C_i} (\Delta M_i) \left( \Delta_\gamma^q D_i \right) P_i. \tag{1}$$

The ratio for biodiversity protection policy $i$ ($R_i$) depends on five elements. Initially there is the ratio of the policy's marginal changes to total economic value ($\Delta TEV_i$) to its marginal costs ($C_i$). That ratio is then multiplied by the resulting change in ecological value ($\Delta M_i$), the change in overall (gamma) species diversity $\left( \Delta_\gamma^q D_i \right)$, and the probability of its successful implementation ($P_i$).

It is useful to point out that over time conservation management practice has converged to using elements similar to this theoretical equation. About 30 years ago, Metrick and Weitzman (1998) critiqued the U.S. governmental spending for endangered species protection because it appeared to be driven by protecting charismatic species; the focus on charismatic species still may be an issue today (e.g., Runge et al. 2019; Bennett et al. 2015). As Possingham et al. (2002) emphasized, another common biodiversity protection practice was for managers to select single species based primarily upon their endangered status on such listings as the IUCN Red List (International Union for Conservation of Nature 2019), Australia's listing under the Environmental Protection and Biodiversity Conservation Act of 1999 (Australian Government, Department of the Environment and Energy n.d.) and the U.S. Fish and Wildlife Services Threatened and Endangered Species Lists (U.S. Fish and Wildlife

Services 2019). Subsequently, costs have added to the assessments because birds and mammals are both commonly protected because of their charismatic nature and they are the most expensive species to protect (Gordon et al. 2019). Boyd et al. (2015) review the use of return on investment analysis as a guide for directing conservation efforts. Joseph and co-authors (Joseph et al. 2008, 2009) broadened the perspective of biodiversity protection practitioners by adding the probability of success to the evaluation of potential programs to create a technique termed "priority threat management." Martin et al. (2018) evaluated that technique from the perspective of maximizing the number of species recovered per dollar invested and Carwardine et al. (2019) reviewed it, paying particular attention to the way it augments traditional analysis with substantial involvement of stakeholders who are chosen to represent notable economic, social, ecological, or knowledge interests in the assessed region. Consequently, even if biodiversity protection managers do not use the specific form of Noah's Equation (1), modern assessment of potential biodiversity protection programs does include the elements of economic and ecological benefits, economic costs, and likelihood of program success that are the core elements of the equation.

I address the impacts of gender concerns related to the components of Equation (1) in the next section. I ignore changes to the overall (gamma) species diversity because they are simply functions of the proportions of each species across the habitat (Tuomisto 2010) and changes in it will not be affected by gender concerns. In some cases, Noah will be a passive observer in the sense that in using Equation (1) he will simply apply the values of those elements that are given to him by society rather than generate them himself. In those cases, the question will be whether Noah recognizes society's gendered biases and responds appropriately to them. In other cases, theory suggests that Noah himself will be the source of a gender bias. In this case, as the decision maker Noah will need to be introspective and self-critical to ensure that he doesn't introduce bias. It will be useful to distinguish between those two cases.

## 3. Analysis of the Gender Biases

### 3.1. Discounting

Although not explicit in Equation (1), the changes in total economic value and a policy's marginal cost are measured in present value terms. If one gender has lower discount rates than the other, then it would have higher valuations in present value terms for the relevant benefits and costs. Thus, it is important to investigate if Noah's gender affects the discounting process.

Recently, Robson and Szentes (2014) concluded that social discount rates (as used for biodiversity protection policies) are lower than private discount rates (used for marketable investments), in part, for biological reasons. This result leads to the question, is there a biological-based gender difference in the choice of discount rates? Drichoutis and Nayga (2015) found that hormonal proxies could not explain the gender differences in time preferences. Later, Fisk et al. (2017) pointed out that the mixed results from studies relating testosterone levels to risk taking (a key component of the discount rate) occur because those studies ignore the confounding relationships between testosterone and social status and between social status and risk taking. Consequently, it seems to be the case that any gender-based differences in social discount rates is not biologically based, but any such difference could be socially based.

Dittrich and Leipold conducted an economic experiment from which they concluded that, "due to evolutionary pressures, women are better able than men to delay gratification and tend to be more self-disciplined" (Dittrich and Leipold 2014, p. 415). Olson et al. (2016) found that the reasons for the lower (than private) social discount rates may differ by gender, with men's choices based upon an intolerance of uncertainty and women's choices based upon empathy. Their conclusion is consistent with both Soutschek et al.'s (2017) finding that women's neural reward systems are more sensitive than men's to prosocial rewards and Zelezny et al.'s (2000) conclusion that women's socialization led them

to have attitudes more favorable to the environment than men have. So, women's discount rates tend to be lower than men's due to their socialization towards empathy and prosocial behavior.

However, one can't forget the "all else constant" assumption. Based upon their literature review Croson and Gneezy (2009) conclude that women are less risk averse than men, which mirrors Olson et al.'s (2016) results. So, women would require a higher risk premium than men, leading them to have higher discount rates.

As I will point out in the next two subsections, women and men face different benefits and costs with respect to biodiversity protection policies. Combining that point with Croson and Gneezy's (2009) finding that women's preferences are more situationally specific than men's implies that the two genders might well view the same situation very differently. Therefore, the applications of empathy–prosocial behavior and of risk aversion will vary more for women than for men. So, the extent that the social discount rates will differ by gender has the potential to vary widely as the social context of the biodiversity policies differ. Even if he is a male who understands the empathy–prosocial behavior and the risk aversion dimensions of gender differentiated discount rates, Noah risks being oblivious to differences between social contexts that change those two dimensions. So, if Noah is utilizing the discounted values as given to him, then he needs to be alert to these aspects so that he does not let societies' biases affect his analysis.

Alternatively, it is possible that as the decision maker Noah might collect the non-discounted values and do the discounting himself. At this point, then, he needs to be aware of his biases: He is less empathetic/pro-social and more risk averse than women are and he will view social contexts differently than women do. It will take careful attention on his part that his choice of discount rate does not affect the assessment of every biodiversity protection project.

### 3.2. Changes in Total Economic Value ($\Delta TEV_i$)

To be clear, the marginal changes to total economic value due to a biodiversity protection policy includes changes to all of the use and the non-use values associated with that policy (Walsh et al. 1984). Regardless of gender, Noah will find the assessment of such marginal benefits difficult to begin with because, as Fisher and Christopher (2007) demonstrated in their oft-cited study, biodiversity hotspots and areas of poverty are strongly interrelated. In particular, Falk and Hermle (2018) found that gendered differences in preferences for altruism, risk taking, and patience increased with economic development, consistent with the theory that greater access to resources allows people more freedom to express individual preferences. So, if the lack of development discourages the expression of preferences but not the creation of preferences (whether gendered or not), then Noah is unlikely to observe the true species valuations when considering many biodiversity protection policies. In addition, if those preferences are gendered then Noah is not likely to observe the distinction because neither genders' preferences would be expressed in the many high-biodiversity, less-developed regions.

Further, there is the issue of whether the marginal values that Noah does observe include both genders' values. This question follows from the existing research that finds that women and men have differentiated opportunities to express environmental values (e.g., Tindall et al. 2003; Hunter et al. 2004; O'Shaughnessy and Kennedy 2010; Xiao and Hong 2010). This finding likely holds for the valuation of species protection in the contexts when men and women face social constraints to their behaviors and contexts for expressing their values. For example, Allendorf and Allendorf (2012) document that women in Nepal focus on their direct relationships with the nearby protected areas when assessing their attitudes towards those areas because that is how they have been constrained to interact with those areas. Further, since their gathering of fuelwood and fodder is both illegal and not marketed, Noah would not be able to observe the values of those female private activities in contrast to the more public male activities.

And, there appear to be some dimensions of gender differences for various species that are not well explored. For example, Martino (2008) found that women from an urban area in Uruguay more highly valued a protected biosphere's role in preserving endangered species than men did, and in

their review article Yang et al. (2018) similarly noted that women tended to value biodiversity and habitat protection more than men did. In contrast to those authors, Ressurreição et al. (2011) found that male residents of and visitors to the Azores had a higher willingness to pay for marine biodiversity than women did. Further, the distinction between species could be gender differentiated. Olive (2012) found that her sample of U.S. women were more concerned about the viability of a tortoise than a snake's viability and Shapiro et al. (2017) found that boys in the US and Bahamas preferred fish and lizards while girls from those countries preferred rabbits and horses. Thus, the lack of generalizable evidence implies that Noah would not know how to factor gender differentiation into his assessment of the value of particular species.

It is clear that Noah the decision maker is not introducing gender bias into the assessment of gains in total economic value himself. Rather, the risk is that he ignores both the known gender biases related to valuation and the uncertainty in the role of gender biases in other valuation contexts.

### 3.3. Policy Marginal Costs ($C_i$)

Khumalo and Yung (2015) nicely identified and classified the various costs associated with the human-wildlife conflict resulting from the protection of wildlife, which allowed them to identify important gender differentiation in the imposition of those costs. For example, they found that Namibian women did not focus on the costs associated with livestock predation because men dealt with livestock, but women were concerned about the loss of field crops because they fed their families. Similarly, Costa et al. (2017) found that women in Guineas-Bissau were more worried than men about crops damaged by wildlife because socially they were expected to feed their families.

The point that the gender differentiation of social roles leads to gender differentiation in cost assessment is crucial to assessing Noah's cost estimates for reasons mirroring the discussing of the benefits from protecting species. Continuing with the example of a policy of protecting wildlife, consider that livestock are likely traded in open markets and/or provisions are made to compensate for livestock losses so the cost of predated livestock can be estimated and counted. But, if crops grown in a garden and used to feed a family are not marketed and/or not compensated for if damaged by wildlife, then those losses will likely not be estimated and counted. So, the extent to which gender differentiates who is responsible for livestock and who is responsible for gardens determines whether Noah's measurements of the costs of protecting wildlife are underestimated for one gender in this context.

Further, Noah also needs to consider two additional themes arising from linkage of biodiversity hotspots and poverty (Fisher and Christopher 2007). First, even within those low income societies the primary costs of biodiversity protection tends to fall upon the relatively poor people (Ferraro 2002; Adams et al. 2004; Sunderlin et al. 2008; Turner et al. 2012) and, second, there are often income disparities between women and men in those countries (e.g., Alvarez-Castillo and Feinholz 2006). Given that the measurement of an economic cost to a person is limited to the entirety of that person's income, Noah's measurements of the costs of biodiversity protection in the less-developed biodiversity hotspots will be limited by the low incomes of the affected women.

A second theme to the linkage between biodiversity and poverty arises because it appears that poor people make economic decisions differently than other people do (see the review article by Adamkovič and Martončik (2017)). For example, Shah et al. (2012) show that poor people tend to borrow more and save less than other people do. Similarly, Haushofer and Fehr (2014) discuss how the stress and negative affective states that poverty causes may induce people to make more short-sighted decisions than otherwise. So, the economically rational Noah will assess the marginal costs of biodiversity protection differently than poor people will, which will only exacerbate his inaccurate assessment of the gender differentiated policy costs.

As with the case of assessing changes in total economic value, Noah risks using the gender biased marginal costs estimates society generates. Slightly different from that case however, in his role of decision maker Noah may exacerbate those biases by thinking rationally rather than as the poor people typically enmeshed in the biodiversity protection policies think.

### 3.4. Changes in Ecological Value ($\Delta M_i$)

While the Abrahamic Noah had divine guidance in loading the Ark, the decision maker Noah must operate in a world of limited ecological information. For example, while Mainwaring (2001) primarily critiqued the common assumption that the value of species increases as the genetic diversity between them increases, two of his other critiques of rules like Equation (1) were information based. First, if we insist on using a genetic distance criterion to measure value then we need to understand that we know only a few of the many genetic distance measures Noah would need to know. Second, we have a similar ignorance in our understanding of the complexity that exists in the ecosystem, such as predator-prey relationships and how multiple species interact when competing for the same resource. The decision maker Noah must wrestle with these same points of ecological ignorance, as well as others.

A key question with respect to such ignorance is: does gender affect what ecological knowledge we have and what ecological knowledge we do not have? McGuire et al. (2012) document that gender-based barriers still persist for female ecologists, which might suggest that questions of greater interest to males than to females are more likely to be studied. Fox and Paine (2019) conclude that papers authored by women have lower acceptance rates for six ecology and evolution journals than papers authored by men, and that the female-authored papers in those six journals are cited less frequently than male-authored papers. So, the ecological research conducted by women might be less well-known than the ecological research by men. Consequently, Noah does need to be aware of the likely social influences determining ecological learning and its dissemination.

Perry (2010) defined the change in ecological value as the ratio of the square root of the number of species affected by the functional group protected by the policy divided by the number of species in that protected functional group. The more species affected by the protected group and/or the rarer that protected group, the more valuable that protected group is. While the denominator of ecological value is simply a matter of counting species, it is the assessment of the ecological functional group during which Noah needs to be concerned about society's existing gender biases.

Gagic et al. highlighted the key issue facing Noah in defining which other species are affected by the protected species:

> Current knowledge of the role of species richness for ecosystem functioning is mainly based on small-scale experiments ... Our findings indicate that we need to integrate the abundance and distribution not only of species, but also of their trait levels within the community to better understand [biodiversity and ecological functioning] relationships in terrestrial animal communities. (Gagic et al. 2015, p. 6)

In short, determining the ecological functioning of species often takes repeated observations over lengths of time in the relevant context.

This is the type of observation that comes naturally as part of the lives of many rural people. However, gender differentiated agricultural roles have been reported broadly, including in South Asia by Agarwal (2000), Upadhyay (2005), and Narayanan and Kumar (2007); in West Asia by Abdelali-Martini et al. (2008); in Mexico by Rimarachin Rimarachin Cabrera et al. (2001) and Chambers and Chambers and Momsen (2007); and in Africa by Mackenzie (2003) and Howard and Nabanoga (2007). Similarly, Allendorf has documented gender differentiated ecological knowledge with respect to tigers in Nepal (Carter and Allendorf 2016) and a nature reserve in China (Allendorf and Yang 2017) resulting from gender differentiated roles.

So, the assessment of the ecological functioning of a habitat will vary by who Noah asks. Naturally the concern is that gender differentiated decision making and the distinction between the private and public spheres in terms of expressing values could lead Noah to omit women's ecological knowledge sets from his assessment of ecological functioning. This is clearly a case in which Noah the decision maker will need to be alert to society's gender biases, so he does not let them distort his application of Equation (1).



As a male himself, however, Noah may also bias the assessment of the ecological value gained by various protection policies. As noted earlier in Section 3.1, Croson and Gneezy (2009) and Olson et al. (2016) conclude that women are less risk averse than men. This is important because Perry and Shankar (2017) show in their Noah model that the ranking of species is directly affected by the decision maker's degree of risk aversion. More risk averse decision makers, females in this case, potentially prefer redundant species, non-charismatic keystone species, decomposers, and primary producers more than Noah would. This is an important case in which theory suggests that Noah could directly bias the analysis in his role as the decision maker.

### 3.5. Probability of Successful Policy Implementation ($P_i$)

As Agarwal (2009) emphasized and others have affirmed (e.g., Rinkus et al. 2017), it can be difficult for many reasons for women to participate effectively in programs that relate to biodiversity management due to gender differentiated access to and agency with biodiversity management. These findings are consistent with the conclusion that societies that are historically plough-positive (growing teff, wheat, barley, rye, and wet rice) tend to be less conducive to women's civic participation than other societies (Alesina et al. 2013). Further, when biodiversity protection must occur outside of protected areas, the gendered patterns of land ownership, as Kieran et al. (2017) found in Bangladesh, Tajikistan, Timor-Leste, and Vietnam, and Dokken (2015) found in Ethiopia, will affect women's ability to protect biodiversity. Even when land ownership is not gendered, the perceptions of husbands and wives of their own and their spouse's participation in farm decision making can be gendered (Twyman et al. 2015).

The constrained ability of women to participate in biodiversity protection is important for Noah because meaningful participation by women tends to increase the probability that the biodiversity protection programs are effective. Agarwal (2000) argues that this increase in effectiveness occurs because women's social networks operate differently than men's networks do, particularly in ways that create social conditions for protecting species. As one example, Pillai and Suchintha (2006) discussed how women organized self-help groups to manage the Periyar Tiger Reserve, including voluntary patrols of the reserve to better maintain its ecological conditions. As another example, Padmanabhan (2008) showed how a community's collective action was more effective than a government program in protecting agrobiodiversity in Wayanad, Kerala, because the collective action built upon women's emphasis on reputation, trust, and reciprocity. More generally, Westermann et al. (2005) concluded that women tend to enhance the collaboration, conflict management ability, and group maturity of natural resource management groups in Asia, Africa, and South America.

Like the earlier situations of Noah observing society's gender biased valuations, Noah is not inserting additional gender bias in his role as decision maker. Instead, the key question for Noah is whether he recognizes and appropriately values the participation of women in biodiversity protection policies. All else constant, he should be favoring those policies that give women meaningful roles. Worryingly, the literature has not asked the question: If women's roles occur in the private sphere (instead of the public sphere) and poverty constrains the expression of values, would an outside observer like Noah actually observe the contributions women make to biodiversity protection?

## 4. Discussion

At this is the point in the note, one would expect a table with one row for each of the components of Noah's Equation (1) and a column with plusses and minuses indicating whether Noah's gender increases or decreases the ranking of a policy and a concluding row giving a nice summary answer. But, analysts have long recognized that the gender-based analysis of biodiversity is complex (e.g., Rocheleau 1995) and they continue to find that one cannot conclude a priori that one gender is more resource-conserving than the other (Meinzen-Dick et al. 2014). So, the fact that combining the gender implications for each of the five components leads me to conclude that gender affects Noah's ranking of species protection policies ambiguously is not surprising.

In one sense, my rationale for that ambiguity is similar to others' conclusions that the complexity in the gender-biodiversity relationship arises from the gender differentiated roles in agriculture and activities related to biodiversity (Pfeiffer and Butz 2005), and the gendered knowledge systems, behavioral expectations, access to resources and institutions (Fortnam et al. 2019). However, instead of simply focusing on Noah's final ranking, I take Arora-Jonsson's point (Arora-Jonsson 2014, p. 12) to heart and assess how gender separately affects each component of his decision rule. Because society's gender differentiation of social roles and the gendered differentiation in the ability to express values affect discounting, marginal benefits, marginal costs, changes in ecological value, and the probability of policy success differently, Noah's ranking of species protection policies could swing wildly after adjusting for those social biases appropriately. Further, as discussed in Section 3.1, if Noah is doing the discounting in his role as the decision maker rather than accepting society's discounted values as inputs, then his own socialization would bias him towards being both less risk averse and less empathetic/prosocial than a female would be. The resulting impacts on the discount rate are also ambiguous and, as discussed in Section 3.4, his socialization in favor of less risk aversion than females could also affect his preferences for species. So, both society's existing biases that affect the inputs Noah receives and his own socialization as a male will affect Noah's ranking of biodiversity protection policies.

Consider the following stylist example of a wildlife protection policy that creates an ecological reserve, with each aspect being consistent with a study discussed earlier. If the household gains electricity because the newly arriving tourists want it, the value that women see in their children being able read at night may not be measured so Noah would underestimate the benefits. If the animals trample the food crops, those costs might be borne by the women and so underestimated by the male. The discount rate women use for the benefits might be lower than otherwise reflecting their greater patience than men, but it might be higher than otherwise for the costs because they are more risk averse than men. The men might have spent more time in the area that is now the ecological reserve and so have a better overall understanding than the women of its ecological functionality. However, it might also be the case that the time women spend in the reserve is spent differently than men's time so they observe different ecological functions than men do, and those functions would be omitted from the analysis. Finally, Noah has to both observe and appreciate the benefits of any participation by women in managing the ecological reserve to accurately access the probability that the reserve will successfully protect the relevant biodiversity. My point is that it is important consider the implications of gender for each of those aspects instead of simply looking at a "grand total" effect.

Of course, implicitly I have been arguing for equality between the genders as the measure of a just society. On the other hand, Duflo (Duflo 2012, p. 1076) concluded, "In order to bring about equity between men and women, in my view a very desirable goal in and of itself, it will be necessary to continue to take policy actions that favor women at the expense of men, and it may be necessary to continue doing so for a very long time." From this perspective, Noah should adjust ranking equation, perhaps by giving greater weight to either the benefits to or the costs incurred by women. This would allow him to take what Duflo might view as a step in the direction of long-term gender equality.

To make matters even more interesting (worse?) for Noah, Grand et al. (2017) point out that many conservation professionals do not even consider cost-effectiveness to be the most important feature when designing a conservation policy. So, despite the previously discussed (in Section 2) convergence of biodiversity protection program assessment methodologies to emphasize same elements as Noah's Equation (1), many conservation professionals would use factors other than or in addition to the key economic components of the equation. And going one step further, as Martinez-Alier (2002) stressed, different social groups can use different, non-economic valuation methods for the same resource at the same time and even the same group can use different methods to serve different purposes. Further, he would broaden the analysis from gender-based differences in value to considering the unrecognized differences related to race, indigenous culture, and other aspects related to socially marginalized

groups. So, it is important for Noah to realize that he does not operate in a vacuum, let alone as a deity-mandated economic judge of what species survive the flood.

**Funding:** I conducted the initial research for this paper as a Fulbright-Nehru Research Scholar sponsored by the U.S Indian Educational Foundation at the Institute of Economic Growth (IEG) at the University of Delhi, which made an earlier version of this paper available as IEG Working Paper #324.

**Acknowledgments:** I thank all of the colleagues who offered constructive comments on drafts at the *II International Conference on Gender Relations in Developing Societies: A 21st Century Perspective* held at Maharaja Agrasen College (Delhi), a seminar at IEG, and a seminar organized by the Zoological Society of Bangladesh at the Zoology Department at Dhaka University. I also wish to thank an anonymous reviewer for providing useful comments about the penultimate draft.

**Conflicts of Interest:** The author declares no conflict of interest. The funders had no role in the design of the study; in the collection, analyses, or interpretation of data; in the writing of the manuscript, or in the decision to publish the results.

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
