# Peer review of "Gender Concerns When Noah the Economist Ranks Biodiversity Protection Policies"

_socsci, doi:10.3390/socsci8100282_

Round 1

Reviewer 1 Report

The author explains that the priority ranking of species, as formulated by the late Martin Weitzman and others, is gendered. In the final paragraph, it is also suggested that the ranking equation is determined by culture and class. I do not disagree with the theses but I would like the author to refine the paper by distinguishing between theory and practice. Many of the points raised do not operate at the theoretical level so the criticism of the ranking equation can be countered. The points that do operate at the theoretical level would benefit from the distinction.

In theory, the total economic value, ecological importance, the probability of success and costs are not gendered. Fundamentally, for example, the total economic value represents society’s views on the value of species or a conservation program. As such, the value includes in equal measure the views of all genders, classes and cultures. Thus, the ranking equation itself is not at issue. What is at issue is the operation of the ranking equation.

The author is correct that in all likelihood, the voice of woman and some other groups will be marginalized because they have less presence in the decision making process. I think the point about woman having different exposure to, and therefore perspective on, the importance of different ecological functions is well made. However, again, at the theoretical level, ecological importance involves the perspectives of all genders and priorities are simply a matter of applying the formula for ecological importance. In practice, the voices of woman may well be marginalized but this speaks to the operation of the ranking equation. In fact, Metrick and Weitzman’s (1998) paper comparing the ranking equation from Weitzman’s (1998) seminal paper to the actual choices made by the Fish and Wildlife Service in the U.S. makes the point that the theory suggests one thing and the operation of the U.S. Endangered Species Act 1973 suggests another. Ecological importance is also often overlooked in priorities in favor of charismatic species.  

In contrast, the author’s thesis does apply at the theoretical level when it is recognized that willingness to pay depends on ability to pay. To the extent that woman have lower incomes than men, their ability to pay, and therefore the size of their vote on the value of species will be less significant. This point also applies to different cultures and the poor in society. Information is also an issue here with different levels and forms of information affecting preferences and therefore impacting upon willingness to pay. Amongst others, this point has also been made by Mainwaring (2001), which would be worth reviewing and citing, and it also applies at a more general level to cost-benefit analysis which is at the foundation of the ranking equation. A second area where the author’s critique applies at the theoretical level is in regards to risk preferences. The author mentions differences in risk aversion in line 109 and a recent paper has highlighted that Noah’s risk preferences matter for prioritization (Perry and Shankar, 2017). By making the distinction between theory and practice, these theoretical points become stronger and the other points that apply to practice are no less important but are clearly delineated as practice-based issues.   

As the ranking equation is theoretical in nature and hardly ever applied in practice, it may be interesting to dedicate some space in the article to policy expressions of the Noah’s Ark problem. For example, the way in which the US Fish and Wildlife Service allocates funds could be analyzed. Similarly, the US Endangered Species Act 1973 has priorities built in that bias against certain species. The Australian state of New South Wales applies the project prioritization approach developed in Joseph et al. (2008), which also should be cited in the paper. This policy could be analyzed to situate the author’s thesis as primarily addressing the operation of the ranking equation.

Fundamentally and theoretically, Noah is a passive decision maker who is simply representing society’s values. As such, the gender of Noah does not matter. It is only when the ranking equation is applied that gender biases will be revealed. The paper would be stronger if this distinction between theory and practice is explained and used to delineate the important points in the paper.  

References

Joseph, L.N., Maloney, R.F. and Possingham, H.P. 2008. “Optimal Allocation of Resources among Threatened Species: A Project Prioritization Protocol.” Conservation Biology 23(2): 328-38.

Mainwaring, L. 2001. “Biodiversity, Biocomplexity, and the Economics of Genetic Dissimilarity.” Land Economics 77:79-83.

Metrick A. and Weitzman, M.L. 1998. “Conflicts and Choices in Biodiversity Preservation.” Journal of Economic Perspectives 12(3): 21-34.

Perry, N. and Shankar, S. 2017. “The State-Contingent Approach to the Noah’s Ark Problem.” Ecological Economics 134: 65–72.

Weitzman, M.L. 1998. ‘‘The Noah’s Ark Problem.’’ Econometrica 66 (6): 1279-1298.

Reviewer 2 Report

-This paper needs additional experiments, some tables and graphs.

-It needs explain its economic focus. Why it ignores the feminist economics?

-Religious references are not necessary.

-Additional contents: A table of literature review. Articles of Barry Commoner.

Round 2

Reviewer 1 Report

I thank he author for the revised draft and the response to my comments. I believe my points have been adequately addressed and clearly expressed and the paper should be published in my opinion. It is an original contribution to knowledge and the author has highlighted an additional factor that decision makers need to be aware of as they prioritise limited conservation funds. Namely, decision makers need to consider the degree to which their own gender biases impact on priorities or the degree to which systemic gender discrimination impacts upon the information available to them as they implicitly or explicitly apply the prioritization equation.

There are three minor editorial comments I would like to make:

Line 81: "About thirty years ago" should say "About twenty years ago" line 86: the sentence is a little unwieldy and I believe would benefit from removing "such listings as"  Lines 90-94: It is not clear what is meant by "costs have added to the assessments". perhaps this is a typo? Maybe it should be "costs have not been incorporated" because birds and mammals have been prioritised despite the costs. Also, The clause starting with "Boyd, Epanchin-Niell..." should be a new sentence. 

Thank you again for the considered response to my comments. 

Reviewer 2 Report

-This review is better than the first version.

-It needs some text editing according to the journal's format.